# Acute and Long-Lasting Effects of Slow-Paced Breathing on Handball Team Coach’s Match Stress

**DOI:** 10.3390/healthcare11091242

**Published:** 2023-04-26

**Authors:** Zoran Nikolovski, Dario Vrdoljak, Nikola Foretić, Mia Perić, Dora Marić, Charalampos Fountoulakis

**Affiliations:** 1Faculty of Kinesiology, University of Split, 21000 Split, Croatia; 2High Performance Sport Center, Croatian Olympic Committee, 10000 Zagreb, Croatia; 3College for Humanistic Sciences, ICPS, 121 31 Athens, Greece

**Keywords:** stress, hormones, salivary glands, endocrine, autonomic nervous system

## Abstract

Stress was previously defined as a response to any demand for change. These demands are noticeable during sport events, not only in athletes but also in coaches. Therefore, this study aimed to determine the influence of slow-paced breathing (SPB) on acute stress to handball coaches during official matches. One professional handball coach, aged 37 (height, 180 cm; weight, 80 kg), took part in the study. His salivary cortisol (C) concentration and alpha-amylase (AA) activity were measured, and his heart rate (HR) was monitored during six official matches. In the first three matches the SPB training protocol was not followed. From match four to match six, the coach performed SPB training every day and directly before the match. The decrease observed in AA was statistically significant with a large effect size (1.80). The trend of change in HR is visible and similar for matches with and without SPB. However, for SPB matches, there is a lower starting point compared to matches without SPB (89.81 ± 6.26 and 96.62 ± 8.10, respectively). Moreover, values of AA on SPB matches were smaller before the match (93.92 ± 15.89) compared to the same time point in non-SPB matches (115.30 ± 26.00). For AA, there is a significant effect size in the half-time (2.00) and after the match (−2.14). SPB matches showed a lower increase in AA. SPB might be used as one of the possible tools that could help coaches in achieving a desirable mental state during the match.

## 1. Introduction

Stress was previously defined as a non-specific response to any demand for change, or as “the rate of wear and tear on the body” [1]. It is usually characterized as individual and personal, which differs among given types of tasks [2]. Stress is connected to negative emotions, and therefore, influenced by two major components. Specifically, cause (e.g., major events or daily hassles) and effect (e.g., subjective response, divided between stress appraisal and emotional response) of stress [3,4,5]. Furthermore, some of the most stressful environments are perceived by the sports participants. Both coaches and athletes are exposed to competitive stress, which is defined as an ongoing transaction between an individual and the environmental demands. Competitive stress is, primarily and directly, associated with competitive performance. It is influenced by the competitive stressors (environmental demands) and competitive strains (an individual’s negative psychological, physical, and behavioral responses to competitive stressors). These factors are the main cause of competitive anxiety and may lead to a negative emotional response to competitive stressors [6]. According to previous studies, athletes have reported that their sensation of stress has been influenced by performance-related, intra- and interpersonal, self-presentational, and organizational issues [7,8]. On the other hand, sports coaches operate within complex, changing environments of matches and deal with athletes and their own stress reactions simultaneously. Specifically, coaches lead athletes to improve their physical, technical, mental, and tactical skills, as well as facilitate their personal and social development. They also contribute to wider societal agendas, such as improving health, youth engagement, social inclusion, lifelong learning, and community regeneration [9]. Furthermore, this complex stressful environment in which coaches exist could lead to burnout [10,11]. This state could lead athletes to perceive such coaches negatively [10,11]. Additionally, there are significant potential health costs of the psychological stress experienced by coaches [9].

The performance of a coach and its effectiveness is often connected with the efficiency and success of his athletes [12]. As previously stated, coaches must accomplish multiple roles and deal with technical, physical, organizational, and psychological challenges in their jobs. It is not surprising that because of high demands, coaches confront various levels of stress [13]. Therefore, coping with stress is an important factor leading to success for coaches [14,15]. According to previous studies, there are several ways of stress management leading to mitigating negative effects that might impact performance [16,17,18]. Strategies that are easy to practice and proven to be useful are breathing techniques [19]. Specifically, diaphragmatic breathing has been widely incorporated in routines for general stress reduction, due to its effect on the parasympathetic nervous system. Lowering blood pressure and HR, a simultaneous increase in peripheral skin temperature, and an overall sense of relaxation are some of its effects [20]. Moreover, studies have shown that slow-paced breathing (SPB) could also be a useful tool in managing stress [21,22]. SPB has positive effects on physiological indicators of arousal and vagal flow, including changes in cardiac rhythm, HR and HRV [21,22,23]. Thus, some authors believe that breathing strategies may be useful in managing stress effectively, for both athletes and coaches [24]. However, a vast majority of studies use different measurement methods, such as psychological measures (questionnaires) [25,26], heart rate (HR), breathing frequency [27,28], and biological markers (e.g., cortisol and alpha-amylase) [29].

The most commonly used methods for monitoring cognitive load and stress in athletes are questionnaires [30]. These questionnaires often refer to the anxiety levels of sports’ participants and examine the influence of the match as the main stressor. The questionnaire used in a study by Arruda, Aoki, Paludo and Moreira [31] had a high correlation with the difficulty of the match. Apart from the subjective indicators, heart rate variability (HRV) showed to be a good objective measure of stress. HRV is defined as the ability of the heart to produce fluctuations in the beat-to-beat interval in response to different situations [32]. It was previously shown that HRV variables change depending on stress and can distinguish between different stressors [33]. In terms of biomarker monitoring, cortisol (C) reflects the activation of the hypothalamic–pituitary–adrenocortical axis, which is involved in the physiological stress response [34]. C is frequently used as a biomarker of the physical and mental (psychological) stress-related functioning of athletes [35]. Alpha-amylase (AA) is an enzyme that catalyzes the hydrolysis of starch into smaller carbohydrate molecules, such as maltose [36]. Additionally, AA reflects the activity of the sympathetic nervous system. It has been shown that physical activity consistently increases AA activity and concentration [37]. Moreover, levels of AA are highly related to an increase in noradrenalin, and consequently, reflect the state of arousal [38]. Biological markers have also been used as indicators of stress caused by sports activity [31,39]. For example, Moreira, McGuigan [35] showed an increase in C and AA regarding mental exertion during physical activity.

Hunt, Rushton, Shenberger and Murayama [40] compared diaphragmatic breathing (DB) and progressive muscle relaxation techniques. The results showed that DB is easy to apply and is an effective tool for athletes to cope with stressors. According to the authors, during DB a lower rate of respiration, higher tidal volume, lower HR, and higher HRV were observed. Additionally, there is evidence that SPB has a positive effect regarding adaptation to psychological stress before and after physical exertion [41]. According to Thelwell, Weston and Greenlees [42], world-class coaches reported relaxation breathing as a rarely used stress-coping strategy. However, the literature review showed good results of stress management in sports while practicing breathing exercises [40,41]. Therefore, the aim of this study was to determine the possible influence of SPB on acute stress in handball coaches during official matches. The authors hypothesize that SPB could have an effect on stress biomarkers and HR. Moreover, if proven to be effective, SPB can be used by handball coaches as a stress coping practice.

## 2. Materials and Methods

### 2.1. Participant

One professional EHF master handball coach (kinesiology professor), aged 37 (height, 180 cm; weight, 80 kg), with 19 years of coaching experience in handball, participated in the study. The participant’s stress was measured in 6 official matches, during which a total of 33 saliva samples were collected. During the course of the study, the subject coached a club in the first Qatar Handball League, which was a cup finalist, and at the end of the season achieved 8th place. The participant was informed about the procedures and aims of the research. He signed an inform consent form and participated voluntarily. All experimental procedures were completed following the declaration of Helsinki (2008), and they were approved by the corresponding authors’ institutional research ethics board (Ethics Board Approval No. 2181-205-02-05-18-002).

### 2.2. Procedures

Salivary C concentration and AA activity were measured, and the HR was monitored. Samples for stress markers (HR, C, and AA) were collected during one rest day to establish the baseline circadian rhythm. Additionally, samples from six official matches in the first Qatar Handball League were collected too. To establish the baseline circadian rhythm of biomarkers, the coach spent the day without physical activity, fasting in his apartment. During the course of the day, from 9:00 a.m. to 9:30 p.m., 10 saliva samples were collected, and his HR was monitored at the same time points [38]. Baseline values corresponded to the same time of sample collection during the matches. Due to the circadian variability of the measured markers, values varied during the day. Prior to collecting saliva samples, the subject stayed still (sitting) for 10 min, and during that period, his HR was continuously monitored (the average values were taken as a result).

In the second half of the study, stress markers were tracked before, during, and after the handball matches (Figure 1). The coached team won three out of six matches (1st match, 34 to 29; 2nd match, 30 to 34; 3rd match, 32 to 28; 4th match, 27 to 30; 5th match, 23 to 30; and 6th match, 34 to 31). All six matches were held in the same hall (QHA sports hall), with 5000 seating places; the dimension of the court was 20 × 40 m, and the temperature ranged between 23 and 25 °C. The matches took place during a period of 35 days, and were scheduled between 4:30 p.m. and 8:30 p.m.

### 2.3. Design

In the course of the first three matches, SPB training or any other stress-relieving protocol was not performed. In the rest of the text, these matches are labeled as NOSPB matches. From match four to match six, the coach performed SPB training twice a day. To assess the SPB breathing frequency profile, the Biograph Infiniti version 6.0.4 was used. The subject entered the mindroom–special room for psychological testing, where the researcher gave him guidelines on how to practice diaphragmatic/abdominal breathing. To guarantee proper abdominal breathing, the participant practiced it for some time (i.e., using diaphragmatic breathing). Inhalation happened through the nose, whereas exhalation happened through pursed lips. A report of testing was generated when the test was complete, including the coach’s performance across all four primary HRV measures (i.e., heart rate (Max–Min), pNN50, SDNN, LF percent of total power) and whether he was able to properly follow the breathing pacer. The participant was asked to complete a low-breathing test, following the pacer on the screens, and providing feedback to the researcher about whether he felt comfortable with the pace at the end of the test to ensure that the calculated breathing frequency was indeed his individualized (resonant) breathing frequency. Our subject took 4.5 complete breaths per minute, with inhalations lasting 4.5 seconds and exhalations lasting 9.1 seconds. The subject was told to download a free breathing app (paced breathing app, Trex LLC, Greenbelt, Maryland USA), which is only a breath pacer, and install it on his smartphone. The participant was instructed to undertake a 20 min exercise on his mobile app at home and 30 min before the match in the coach’s room. Prior to the study of immediate influence of SPB on stress on the handball coach during matches, the participant practiced SPB training during a 25-day period, twice a day, with a total of 920 min of slow-paced breathing in 46 sessions.

### 2.4. Instruments

Saliva samples were taken at 5 collection points: 30 min before the match (BM), at halftime (HT), directly after the match (EM), and 45 (45) and 90 (90) min after the match; for the matches with SPB, one more point is included: 20 minutes before the match (SPBBM) (Figure 1). The coach avoided eating a major meal an hour before sample collection and rinsed his mouth thoroughly with water 10 min before each sample collection. For this purpose, SalivaBio Oral Swabs-SOS (Salimetrics LLC, State College, PA, USA) were used; they were placed underneath the tongue on the floor of the mouth for 2 min. After collection, the swabs were placed into a storage tube and immediately refrigerated. Within 2 h of sampling, the samples were frozen at below −20 °C until centrifugation. On the day of the analysis, the samples were completely thawed and centrifuged at 1500× *g* for 15 min. After centrifugation, assays were performed. Saliva cortisol and alpha-amylase were analyzed with a commercially available enzyme-linked immunosorbent assay (ELISA) purchased from Salimetrics LLC (State College, PA, USA) on a microplate reader (Infinite 200PRO, Tecan, Mannendorf, Switzerland). All samples were analyzed in the same batch to avoid intra-assay variability.

HR was measured using a heart rate monitor (Polar M430, Polar Electro, Kempele, Finland) that the coach wore for a total of 4 h: 30 min before the match, throughout the match, and 90 min after it (Figure 1). 

### 2.5. Statistics

The non-parametric/parametric nature of the variables was tested using the Kolmogorov–Smirnov test procedure. The calculation of the descriptive statistic parameters included means, standard deviations, and percentages (for HR and biomarkers values). The differences among the measurements of salivary biomarkers were calculated by the magnitude-based Cohen’s effect size (ES) statistic, with modified qualitative descriptors (trivial ES: <0.2; small ES: 0.21–0.60; moderate ES: 0.61–1.20; large ES: 1.21–1.99; and very large ES: >2.0) [41]. The percentage of change was calculated accordingly and presented as percentage values. Firstly, we deducted two measurements between biomarkers (e.g., C before the activity and in the middle of the activity). Secondly, the first measurement was divided with the difference obtained by the previous step (e.g., C before activity). For example,
(1)$$Difference=C\left(atthemiddleoftheactivity\right)-C\left(beforetheactivity\right)\;Percentageofchange=\frac{Difference}{C\left(beforetheactivity\right)}x100$$

Statistica ver. 13.0 (Dell Inc., Austin, TX, USA) was used for the analyses, and a *p*-level of 95% (*p* < 0.05) was applied.

## 3. Results

### 3.1. Acute Effect of Slow-Paced Breathing on HR, C, and AA before the Matches When SPB Was Performed

Figure 1 shows the acute influence of SPB on biomarker values at BM and SPBBM time points. There was a clear decrease in HR (89.81 ± 6.26) and AA (61.47 ± 31.51) comparing the values before and after (86.00 ± 10.30; 21.90 ± 3.84, respectively) SPB. A statistically significant decrease was observed in AA with a large effect size (1.80), whereas the values of the HR did not achieve statistical significance. Further statistical analysis showed that C was not influenced by SPB. Regardless, we noticed an increase in C levels after the SPB period, although without statistical significance.

### 3.2. Effects of Slow-Paced Breathing on Heart Rate

The HR dynamics before, during, and after the matches are presented in Figure 2. The results show a trend of dynamics for matches with and without SPB. In both SPB and NOSPB similar trend in HR change was observed. However, there is a visible lower starting point at BM for SPB matches (89.81 ± 6.26) and a smaller increase at HT (93.92 ± 15.89) compared to the matches when SPB was not executed (96.62 ± 8.10 and 115.30 ± 26.00, respectively). Additionally, the differences of HR after the match are noticed in both the 45- and 90-minute measurement points.

### 3.3. Effects of Slow-Paced Breathing on C

Figure 3 shows the dynamics of C and its differences before, during, and after the match. C dynamics did not show a significant effect size in regard to SPB. However, it is visible that the change between the first two points of measurement (BM and HT) differ between NOSPB (BM, 0.40 ± 0.07; HT, 0.44 ± 0.18) and SPB (0.31 ± 0.11; 0.29 ± 0.05, respectively). In the NOSPB matches, the increase in C can be noticed, and in the SPB matches, C is almost constant. The rest of the timeline reveals the decrease in C for both conditions, respectively.

### 3.4. Effects of Slow-Paced Breathing on AA

Alpha-amylase dynamics and the differences of its levels relative to match time points are presented in Figure 4. AA dynamics exhibit a very large effect size in the HT (2.00)(NOSPB, 123.99 ± 10.17; SPB, 86.24 ± 24.66) and 90 (3.50)(31.81 ± 4.63; 46.75 ± 3.87, respectively) measuring point, whereas in the EM (128.67 ± 10.17; 86.24 ± 24.66, respectively), it presented large ES (1.51). Specifically, SPB matches showed a lower increase in AA throughout the whole match. After that, AA had a decreasing trend after the match. However, 90 minutes after the match, lower values in NOSPB matches (31.81 ± 4.63) in comparison to SPB (46.75 ± 3.87) are noticed.

The percentages of change in all measured parameters are presented in Figure 5. The results imply that SPB influenced all biomarkers before and/or throughout the match. HR had the biggest increase in the first two points of measurement. The SPB matches demonstrated a smaller increase (40%) than NOSPB (144%) in AA. Similar results were noticed in C (SPB–8%; NOSP 11%) and HR (SPB 5%; NOSPB 19%). In the second half of the matches, AA exhibit a lesser increase (SPB 6%; NOSP 4%). Additionally, HR experienced a bigger increase in SPB (24%). C had a bigger decrease in NOSPB (−39%) than SPB (−12%).

## 4. Discussion

Precompetitive anxiety and arousal play an important role in coaching at a high level [22]. There is an evident lack of studies researching stress coping tools and stress handling strategies for coaches. This study was carried out in order to find a simple and effective strategy to deal with competitive stress for coaches. Accordingly, we report several important findings: (1) SPB influences all measured stress biomarkers; (2) A long lasting effect of SPB was detected; and (3) HR and AA have similar dynamics under the influence of SPB, acute and long-lasting, while there is no effect of SPB on C level. These findings corroborate the hypothesis that SPB could help coaches mitigate the influence of stressors during handball matches.

As reported previously, a competition setting elicits a higher stress response in coaches compared to trainings. It was speculated that the pre-competitive effect increases stress biomarkers levels (HR, C, and AA) [43]. This effect is an anticipated response to the stressful situations [44,45]. This leads to the triggering of the sympathetic nervous system (verified as an increase in HR and AA) and the hypothalamic–pituitary–adrenal axis (observed as an increase in C) [46]. Altogether, it may induce the development of negative emotional states and the disruption of decision making [47,48]. Therefore, it is of major importance to cope with stressful situations in order to gain the optimal state of arousal and a low level of anxiety before the competition.

According to Hunt, Rushton [40], diaphragmatic breathing (DB) has been shown to be a good technique to cope with stressful situations. Specifically, the authors examined the influence of DB on psychological stress in 76 varsity athletes. DB was compared to progressive relaxation and was shown to be a better tool for stress management. Another study showed how SPB has a positive effect on psychological stress after physical exertion. Hence, it can be noted that SPB has similar effects as DB [41]. The above-mentioned studies have similar findings to our study, where the effect of SPB is evident in all measured biomarkers. Specifically, SPB decreased HR and AA immediately after the breathing sessions (a higher effect size (1.80) is observed in AA), while C increased after the SPB. This finding could be explained by the circadian rhythm of C. While C during measuring drops, AA increases [38].

Blum, Rockstroh and Göritz [49] reported the influence of SPB on HRV. The participants of their study carried out the task after which the SPB was performed, followed by another task. The results showed a decrease in HR after SPB. Additionally, the chronic effects of SPB on HR and vagal tone were examined in other studies. In one study, a single session of SPB (preceded by a familiarization session) was capable of enhancing the vagal tone under cognitive stress [41]. Additionally, the duration of the breathing sessions had an influence on the spontaneous respiratory frequency in the resting measurement. Specifically, the respiratory frequency appears to decrease with the session duration, thus potentially contributing to additional relaxing effects [50]. However, these studies did not examine the long-lasting effects of SPB (completing a single session), and therefore, there is a need for future studies to address this effect in multiple sessions over a longer period of time. These long-lasting effects have been previously shown, but only few parameters were taken into consideration. Blum, Rockstroh and Göritz [49] concluded that regular breathing exercises could influence HRV in the long term. Furthermore, slow-paced breathing appears to be a promising cost-effective technique to improve subjective sleep quality and cardiovascular function during sleep in young healthy individuals [51].

In our study, the long-lasting effect of SPB was demonstrated. As the coach practiced SPB daily, this caused all the measured biomarkers to drop. Specifically, HR and C, at the BM point of measurement, showed lower values in the SPB matches (6.8 bpm, 0.16 μg/dL). This finding suggests that the coach started match preparations at a lower level of stress, which could possibly be a consequence of SPB practice. Hence, our findings might have implications in the sports competition scenario.

Lastly, the effects of SPB could also be noticed through the dynamics of AA and HR, throughout the match. The acute influence of SPB was mentioned previously, as studies reported similar findings to ours [41,49]. Moreover, Kant [45] examined the effects of a relaxation schedule on the arousal levels and performance of college handball players. The results of their study showed the effects of breathing techniques. However, those studies reported the chronic effect on vagal tone, HR, and HRV, whereas in our study, the stress markers C and AA are also included. The dynamics of C is similar for both SPB and NOSPB matches, and show a decrease before, during, and after the match. On the other hand, AA and HR dynamics differs from C. An AA and HR increase was noticed until HT, and then there was a decrease. Specifically, the AA and HR percentage of change for the SPB matches was smaller than in the NOSPB matches. In the BM point, there is even a larger effect size between SPB and NOSPB for AA (1.80).

## 5. Conclusions

In conclusion, the main goal of every sports coach should be the fast and clear understanding of game situations and good decision making. To fulfill such roles, the coach must remain calm, optimally aroused and with as little stressful interferences as possible. Our study introduced SPB as one of the possible tools that could help handball coaches in achieving a desirable mental state during a match. The results indicate that the application of SPB could decrease the level of stress in an acute and long-lasting manner.

The main limitation of this study is the small sample size of the matches used and the monitoring of a single coach. The study failed to assess some very important stress biomarkers that could better explain the handball coach’s emotional and physical stress. The relationships of adrenalin, blood glucose, and breathing frequency with the biomarkers used in our study (HR, AA, and C) had been previously studied and could have given us a better insight into the stress state. The mentioned study limitations limit the generalization of results since the study was only focused on one subject and on an insufficient number of relevant stress biomarkers. Future studies should include more coaches and a higher number of matches. Additionally, monitoring a greater number of biomarkers (e.g., glucose and other hormones connected to stress) could give us more robust and clearer explanations of the gathered data.

## Data Availability

Not available.

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
