# Peer review of "Acute and Long-Lasting Effects of Slow-Paced Breathing on Handball Team Coach’s Match Stress"

_healthcare, 2023, doi:10.3390/healthcare11091242_

Round 1
Reviewer 1 Report
In a manuscript entitled Acute and Long-Lasting Effects of Slow-Paced Breathing on Team Handball: Preliminary Study , the authors examine the influence of slow breathing (SPB) on the acute stress of a handball coach during official matches. Cortisol concentration in saliva and alpha-amylase (AA) activity were measured, and heart rate (HR) was monitored during six official matches. The trial included only one subject. The first three matches were played without SPB training protocol, and in the remaining three the subject performed SPB training every day and immediately before the match. Lower AA values ​​were noted before the SPB match and a smaller increase in AA during the match. The authors state that these differences are statistically significant, but they do not support this with data.
I believe that the results of the work are not supported by sufficient data and that the examined sample is too small to have scientific significance.
Author Response
Dear reviewer,
Thank you for your comments. We have revised and corrected the manuscript accordingly. Therefore, the answers are attached.

Reviewer 2 Report
Dear Authors,
Kindly address the following queries:
1. Salivary C concentration and AA activity were measured, and HR was monitored. Samples for stress markers (HR, C, and AA), were collected during one rest day to establish the baseline circadian rhythm.- Can you let me know why rest has given 1 day?
2. The participant was instructed to go through a 20-minute exercise with his mobile app at home and 30 minutes before the match in the coach’s room. SPB training was practiced during 25 days period, twice a day, with a total of 920 minutes of slowpaced breathing in 46 sessions.? Is there any Reference for this specific training period?
3. Even no Reference is mentioned for the Methodology part described in the manuscript?
4. Whether the BMI of the participant has any influence on the outcome?
Author Response

(The authors gave the same response as above.)

Reviewer 3 Report
This is a novel and very interesting paper.
More justification for all-day fasting [line 104] should be provided. Fasting in itself - or even not eating at an accustomed time - may be stressful for some individuals. A day without physical activity might also be stressful for a sportsman. Does the rest day provide an adequate baseline?
I am surprised that the HRV measure RMSSD was not used. This omission should be justified - or, preferably, rectified.
Asterisks are mentioned in the captions to Figs 2 and 3, but are not included in the Figs themselves.
A reference or justification should be provided for the 'modified qualitative descriptors' for Cohen's ES.
Name and supplier of free breathing app are not provided.
Typo line 134: smartphone. ‘’sThe participant
It would be helpful to know if the authors plan to develop this research themselves [lines 292-7].
Author Response

(The authors gave the same response as above.)

Round 2
Reviewer 1 Report
All requests have been met. I recommend publishing the article in its current form.
Author Response
Dear reviewer,
Thank you for your comments.
Kind regards